# communications
# engineering

# Multiscale coupling of surface temperature with solid diffusion in large lithium-ion pouch cells

Jie Lin [1], Howie N. Chu[1], David A. Howey [1,2] & Charles W. Monroe [1,2✉]

Untangling the relationship between reactions, mass transfer, and temperature within lithium-ion batteries enables approaches to mitigate thermal hot spots and slow degradation. Here, we develop an efficient physics-based three-dimensional model to simulate lock-in thermography experiments, which synchronously record the applied current, cell voltage, and surface-temperature distribution from commercial lithium iron phosphate pouch cells. We extend an earlier streamlined model based on the popular Doyle–Fuller–Newman theory, augmented by a local heat balance. The experimental data reveal significant in-plane temperature non-uniformity during battery charging and discharging, which we rationalize with a multiscale coupling between heat flow and solid-state diffusion, in particular microscopic lithium intercalation within the electrodes. Simulations are exploited to quantify properties, which we validate against a fast full-discharge experiment. Our work suggests the possibility that non-uniform thermal states could offer a window into—and a diagnostic tool for—the microscopic processes underlying battery performance and cycle life.

[1] Department of Engineering Science, University of Oxford, Oxford OX1 3PJ, UK. [2] The Faraday Institution, Harwell Campus, Didcot OX11 0RA, UK. ✉email: charles.monroe@eng.ox.ac.uk

Specific energy, cycle life, safety, and cost of lithium-ion batteries have all substantially improved in the past decade[1–4], but challenges remain for high-power applications. The heat generation that accompanies charge or discharge[5] generally increases both a cell's mean temperature and the extent of its temperature non-uniformity, which can degrade performance and cycle life[6–9]. In worst-case scenarios, non-uniform heating of lithium-ion batteries at high power can facilitate catastrophic thermal runaway[10,11]. Outside of the effect on degradation, local variations of temperature within a battery cell can substantially impact characteristics such as equilibrium voltage and apparent internal resistance[12,13]. Temperature measurements contain rich information about the physicochemical processes that govern battery behaviour. If the microscopic origins of a battery's thermal footprint are understood in sufficient detail, it may be possible to diagnose complex microscopic information from macroscopic temperature measurements. Temperature distributions can be monitored transiently via non-invasive thermal imaging techniques[14–18] and implanted sensors[19].

Battery models are essential tools for exploring how different physical mechanisms contribute to measured behaviour[20,21]. Experimentally validated models are further useful for estimating material properties and optimising cell designs[22]. Physics-based electrochemical simulations based on porous-electrode theory, such as the Doyle–Fuller–Newman model[23–25], are well established. Thermal models are typically coupled to electrochemical models through an energy balance equation that accounts for Joule (ohmic) heating, reaction heat, and entropic heat. Thermal effects were first considered in a zero-dimensional global form by Bernardi et al.[5], and later treated in a local but spatially one-dimensional form by Srinivasan and Wang[26]. Electrochemical–thermal battery models have been modified for various cell geometries and operating conditions[13,27–29]. Most simulations deal with three-dimensional geometries by decoupling the thermal problem from the charge-transport problem, solving a one-dimensional Doyle–Fuller–Newman model normal to the electrode sandwich at a given location, then using the results of that computation to produce a generation term in a homogenised, three-dimensional heat equation[30]. Despite the fact that almost all of the porous-electrode-theory investigations in the literature focus only on the electrochemical response of a single layer in the 'through-plane' direction perpendicular to the current collectors[31–36], the 'in-plane' distribution of current can be equally or perhaps even more important, especially in large-format cells[8,18,37,38]. Practically, both safety and degradation are impacted by temperature non-uniformity. Local hot spots typically have lower resistance than their surroundings, causing the active material there to be stressed more intensively by cycling. Moreover, although the global temperature of a cell may be within safe operating limits, catastrophic failure due to thermal runaway can be induced if these limits are exceeded locally.

This paper shows that the surface-temperature distribution across a large-format cell is an effective probe for battery diagnostics. We investigate non-uniform in-plane temperature distributions during battery charging and discharging, and explore various electrochemical processes that may rationalise them. Through use of a judiciously designed test rig that minimises heating due to external wiring and tab contacts[16], lock-in thermography[39] of four commercial large-format 20 Ah LFP/ Graphite pouch cells from A123 Systems is performed while the cells undergo square-wave cycling or constant-current discharge. The results of these experiments are simulated using a new computationally efficient three-dimensional battery model.

We formally derive the streamlined model previously proposed by Chu et al.[16] from a three-dimensional version of the Doyle–Fuller–Newman model that is extended with a local heat

balance. This process reveals several natural routes to produce reduced-order models that account rigorously for additional phenomena, while retaining the high computational efficiency and parsimonious parameter set of the streamlined model.

Excellent agreement between simulations and experiments is obtained by extending the streamlined model to include solid-phase diffusion dynamics. A single lumped diffusion time for both electrodes suffices to match experimental temperature profiles. The three-dimensional aspect of the electrode model differs substantially from typical approaches based on porous-electrode theory, and provides new insights into the dominant physical mechanisms that result in non-uniform temperature distributions. As well as reducing the large set of unknown material properties involved in Doyle–Fuller–Newman theory to a set of just a few observable parameters, the order reduction makes our model computationally efficient enough to support inverse-modelling algorithms based on iterative solutions of full finite-element simulations. We demonstrate, by extracting parameter values from measured cell data, that the streamlined model with solid diffusion can accurately estimate solid-phase diffusivity, as well as key material properties such as electrolyte conductivity, interfacial exchange current density, specific heat capacity, and cell-reaction entropy, among others.

## Results: temperature non-uniformity and solid-state diffusion

Lock-in thermography experiments were conducted to measure the surface temperatures of pouch cells synchronously with their voltage output under given applied currents. Experimental data were gathered using the test rig depicted in Fig. 1, for which the experimental setup and procedures were established by Chu et al.[16]. A test procedure flow-chart can be found in Fig. S1 of the Supplementary Information. Here we report data from two sources: square-wave-excitation cycling experiments, of which the data sets that cycled around a 30% state of charge (SOC) were reported earlier[16], but the others were not; and full-cell discharge experiments, performed specially for this report. All experiments used 20 Ah pouch cells from A123 Systems, which have a lithium iron phosphate (LFP) positive electrode and a graphite negative electrode.

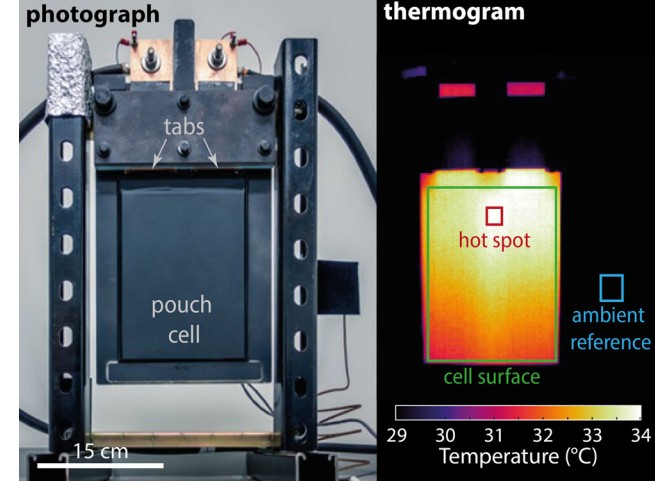

**Fig. 1 Experimental setup.** (photograph) Image of the experimental test rig, showing the pouch cell suspended in a plastic frame, oriented with tabs upward, with wiring connected to the cell via copper bars. (thermogram) An infra-red image of the surface-temperature distribution at an instant during lock-in thermography, showing the hottest pixel and the pixels used to calibrate infra-red temperature.

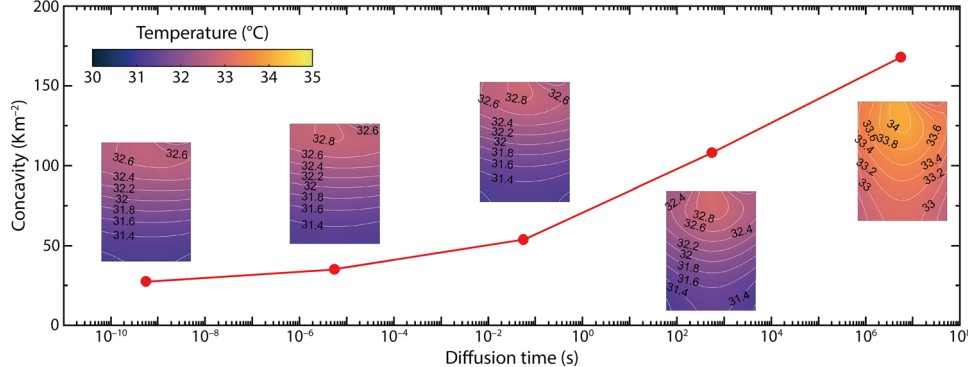

**Fig. 2 Horizontal temperature concavity through the hot spot is controlled by solid-state diffusion time.** Insets: simulated surface-temperature profiles after 2500 s of square-wave cycling with 80 A applied-current amplitude and 100 s period with a 30% initial state of charge (4C-100s@30%), at various diffusion times $t_d$.

**Table 1 Parameter summary. Parameters yielded by model fits based on the new reduced-order model (This work) and the model of Chu et al.[16] (Streamlined model), extracted from experimental lock-in thermography data under 4C-100s square-wave applied currents, with differing initial states of charge.**

| Parameter | Symbol | This work | | | Streamlined model | | |
|---|---|---|---|---|---|---|---|
| | | 30% | 50% | 70% | 30% | 50% | 70% |
| Exchange current density (A cm$^{-3}$) | $ai_0^\theta$ | 1.86 | 1.80 | 1.86 | 143 | 120 | 119 |
| Reaction activation energy (kJ mol$^{-1}$) | $E^\theta$ | 29.5 | 29.2 | 29.7 | 30.8 | 30.6 | 28.8 |
| Effective ionic conductivity (S m$^{-1}$) | $\kappa$ | 0.046 | 0.043 | 0.045 | 0.022 | 0.019 | 0.022 |
| Temperature coefficient of $\kappa$ (mS m$^{-1}$ K$^{-1}$)) | $\alpha$ | 2.4 | 2.7 | 2.4 | 2.0 | 2.0 | 3.0 |
| Open circuit potential gradient (V) | $k_U$ | 0.35 | 0.24 | 0.18 | 0.36 | 0.22 | 0.17 |
| Diffusion time (s) | $t_d = \frac{r_0^2}{D_s}$ | 552 | 590 | 622 | | – | |
| Entropy change (J mol$^{-1}$ K$^{-1}$) | $\Delta S$ | −13.5 | 7.7 | 9.7 | −13.5 | 7.7 | 9.7 |
| Heat transfer velocity (µm s$^{-1}$) | $\frac{h}{C_p}$ | 5.11 | 5.06 | 5.00 | 5.39 | 5.20 | 4.98 |
| Effective thermal conductivity (W m$^{-1}$ K$^{-1}$) | $k$ | 1.1 | 1.2 | 1.1 | 54.4 | 72.0 | 60.1 |

In all cases, lock-in thermography was performed using cells initially equilibrated at ambient temperature. Before each cycling experiment, the cell was discharged from 100% SOC using Coulomb counting to a predetermined initial SOC of 30, 50, or 70%. Cells were cycled galvanostatically, alternating between charge and discharge periods of equal length for the 2500 s duration of the experiment, although the first charge step was performed over a half-period to keep the cell's time-averaged SOC centred at its initial value. The applied current was set at 2C or 4C (i.e., 40 A or 80 A), with periods of 50 or 100 s for one charge/discharge cycle. The cell voltage and ambient temperature were recorded at 1.0 Hz.

Thermograms of the cell surface were captured via a thermal imaging camera, allowing online visual monitoring of cell temperature. A physics-based battery model was solved using COMSOL Multiphysics software to simulate the electrical and thermal responses to the square-wave current excitation (see 'Methods' section for more details). For all the square-wave cycling tests, parameter estimation based on a previously introduced streamlined model[16] could capture the hot-spot, cold-spot, and surface-average temperatures accurately, but failed to predict the correct horizontal temperature distribution on the surface— that is, the distribution across the largest surface of the pouch cells, in the direction perpendicular to the tabs (see Fig. S3 of the Supplementary Information).

In hopes of improving fits of the horizontal temperature distribution, several reduced-order models were derived, each based on the Doyle–Fuller–Newman (DFN) model with an added local heat balance. These produced a variety of extensions to the

streamlined model of Chu et al.[16], as described in Supplemental Note 2. We found that the assumption of linear kinetics in place of nonlinear Butler–Volmer kinetics did not affect observed surface-temperature distributions (see Fig. S4). The inclusion of solid-state diffusion, however, did have a fairly large macroscopic effect. In particular, solid-state diffusivity was found to be the only parameter that had observable impact on the horizontal temperature distribution. This finding suggests that there is a close coupling between thermal transport and solid-state diffusion in the electrode particles, and further, that consideration of this coupling is necessary to account for the horizontal temperature variation in large-format pouch cells.

Figure 2 presents simulation results that demonstrate how the spatial concavity of the horizontal temperature distribution (along a horizontal axis through the hot spot) varies with the solid-state diffusion time constant $t_d$ (cf. Table 1). Details of how this concavity was estimated are provided in Supplemental Note 1 and Fig. S5. The horizontal concavity flattens as the diffusion time constant decreases, ranging from 168 to 35 K m$^{-2}$. At $t_d = 5.6 \times 10^{-6}$ s—an unrealistically fast value[40,41]—the hot spot lies very close to the cell's top edge, and the horizontal temperature variation is minimal across the vast majority of the cell surface. This behaviour qualitatively agrees with results presented by Chu et al.[16], whose model derives from the assumption that solid-state diffusivity is infinitely large, as explained in Supplemental Note 2. If the diffusion time constant increases (i.e., the diffusion coefficient decreases), then horizontal temperature non-uniformity increases. At diffusion time constants above 100 s—values more representative of real electrode materials[40,41]—significant extra

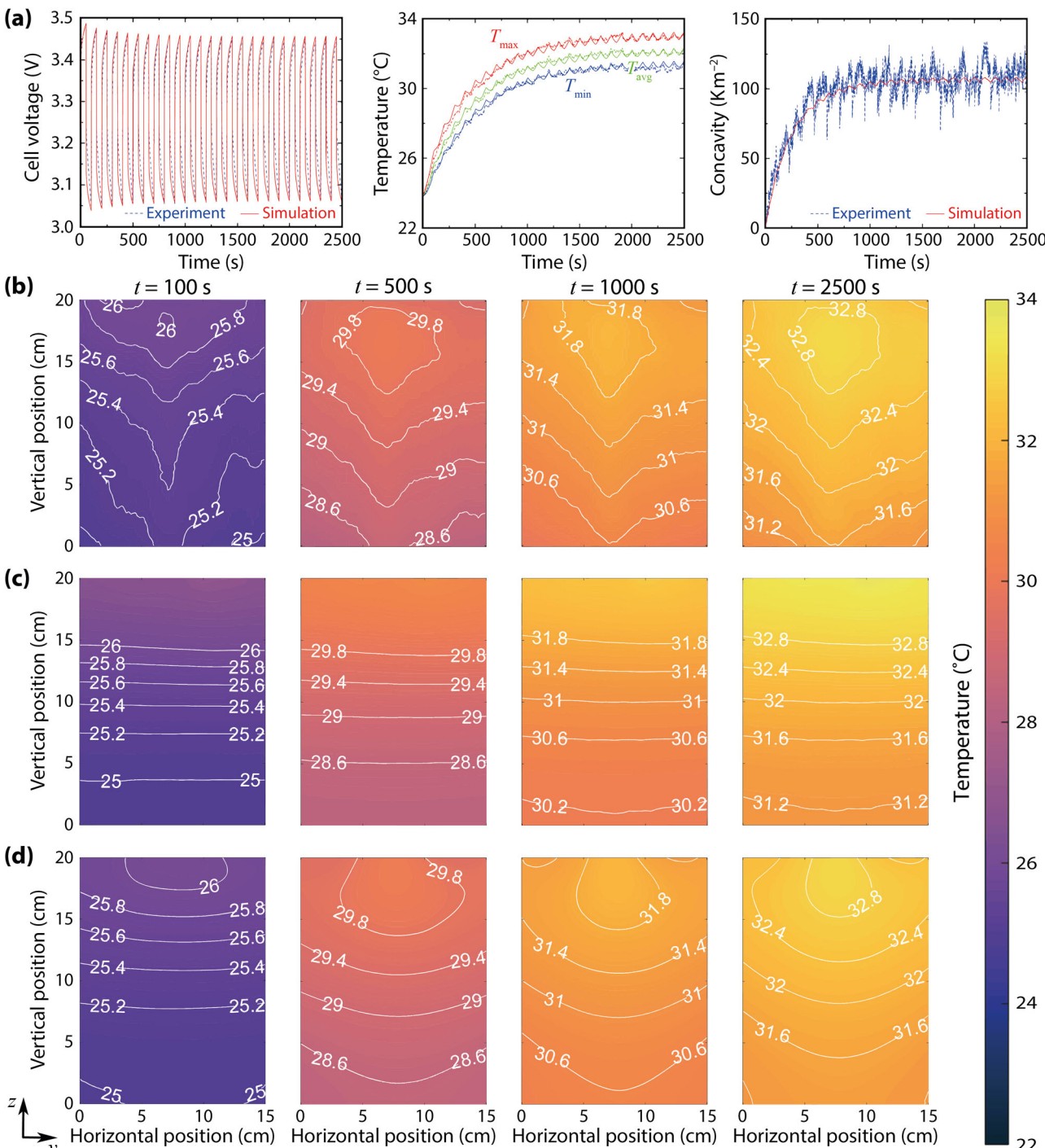

**Fig. 3 A model with solid-state diffusion produces simulations that match experimental surface temperatures.** All data shown are for a 20 Ah LFP pouch cell under 4C-100s square-wave cycling around an initial 30% state of charge. **a** Experimental (dashed) and simulated (solid) cell voltage, temperature and horizontal temperature concavity. **b** Experimental measurements of surface temperature after 100, 500, 1000, and 2500 s of 4C-100 s cycling. **c** Output of the streamlined model by Chu et al.[16], which neglects solid diffusion, trained on the experimental data shown in **a**. **d** Output of the model proposed in this work trained on the data from **a**. The battery tabs (not shown) are on the top edge of the thermal images in **b**–**d**.

heat generation occurs, causing an even larger temperature difference between the central vertical axis of the cell surface and its left or right edges, with generally higher absolute temperatures everywhere.

Typical experimental results for a cell cycled from an initial 30% SOC under a square-wave current having 4C amplitude and 100 s period—abbreviated henceforth as 4C-100s@30%—are shown in Fig. 3 and Supplementary Movie 1. Before $t = 100$ s, the

maximum cell temperature occurs in the areas directly adjacent to the tabs, and a temperature gradient develops primarily in the vertical direction. Near $t = 100$ s, a small 'hot spot'—a point maximum of temperature on the cell surface—appears near the top edge of the cell and begins to move downward along the vertical centre line. Between $t = 100$ s and $t = 500$ s, a domain of higher temperature around the hot spot gradually expands, eventually achieving a relatively stable shape and location when

$t > 500$ s. The average cell temperature also arrives at a state in which it fluctuates around a relatively constant elevated value after about 1500 s. The mean surface temperature in this periodic steady state is determined by the balance of Joule heating and convection from the cell surface, and the temperature fluctuations arise from entropic effects associated with the cell reaction. In the periodic steady state, the high-temperature area around the hot spot generally swells and contracts when applied currents have opposite signs; the hottest area expands when the reaction entropy effect is exergonic, and contracts when it is endergonic.

The strong and relatively isolated correlation between solid-phase diffusion and horizontal temperature concavity (*cf.* Fig. 2) justifies the estimation of solid-state diffusivity with temperature measurements. Taking the concavity of the temperature distributions into account, model parameters were estimated by fitting 4C-100s@30%, 4C-100s@50%, and 4C-100s@70% data using nonlinear least squares, as detailed in the 'Methods' section. The parameters resulting from this estimation process for one cell are listed in Table 1, and the fits for 4C-100s@30% with the same cell are also plotted on Fig. 3a. Corresponding to the experimental thermograms in Fig. 3b–d, Fig. 3c–d show simulation results for 4C-100s@30% using the prior streamlined model[16] and the proposed model, respectively, both based on the best-fit properties provided in Table 1.

Replicate square-wave excitation tests were also performed for three other cells from the same manufacturing lot. Estimated property values from all four cells tested are reported in Table S2. Cell-to-cell variation in the parameters extracted from model fitting was explored using the 4C-100s excitation tests around 50% SOC. Parameter estimates agreed well across cells. The most variable property estimates were open circuit potential (OCP) gradient, with a standard error of 8%, ionic conductivity (7%), and diffusion time (4%); all other parameters agreed within 1% from cell to cell.

It is clear that the model incorporating solid-state diffusion captures more features of the surface-temperature distribution. This is because solid-state diffusion introduces an additional time constant to the system. Two key relaxation times are visible in Fig. 3a besides $t_d$: a reaction relaxation time $t_{rxn}$, and a thermal time constant $t_{th}$. A dimensional analysis shows that

$$t_{rxn} \sim \frac{2RT\bar{Q}}{Fk_U ai_0^\theta \delta} \approx 40 \text{ s} \quad \text{and} \quad t_{th} \sim \frac{\tilde{C}_p \delta}{h} \approx 1400 \text{ s}, \qquad (1)$$

where $R$ is the gas constant, $T$, the ambient temperature, and $F$, Faraday's constant; $\bar{Q}$ is the rated capacity of the cell per unit of superficial electrode area (20 Ah/300 cm²) and $\delta$, the cell thickness (7 mm); the remaining parameters are defined in Table 1. The short-time relaxation of cell voltage is controlled by $t_{rxn}$. This is an electrical time constant, which arises from the fact that the slope of the cell's OCP with respect to its SOC acts like an admittance, and the exchange current density acts like a conductance. Thus the interface relaxes somewhat like a parallel RC circuit, with the OCP gradient providing the (inverse) capacitance, and interfacial charge-transfer kinetics providing the resistance. Notably, this time constant has not been considered in most asymptotic analyses of the Doyle–Fuller–Newman model[42,43]. The thermal relaxation time $t_{th}$ is much longer than the other time constants. It controls the relaxation of the voltage envelope and the average cell temperature. As mentioned earlier, the solid-state diffusion time constant $t_d$ controls how the horizontal temperature concavity relaxes.

Parameter values in Table 1 are similar to those found by fitting with the streamlined model[16], with three notable exceptions—ionic conductivity, exchange current density, and thermal conductivity. Since work by Chu et al.[16] ignored the concentration polarisation caused by solid diffusion, the resulting voltage drop could only be attributed to poor effective ionic conductivity in the electrolyte, which had to be underestimated by *ca.* 50% to fit the cell-voltage response. Reduced ionic conductivity increases the amount of Joule heating, causing an overestimation of exchange current density in order to lower interfacial resistance and match the temperature. The change in thermal conductivity owes in part to the inclusion of hot-spot position in the cost function used during parameter optimisation, as discussed in the 'Methods' section. Separate simulations showed that changing the electrode's effective thermal conductivity tunes the vertical position of the hot spot on the cell surface. This alteration of the cost function led to fitted thermal conductivities of the order of $1$ W m$^{-1}$ K$^{-1}$, placing results in good agreement with independent thermal-characterisation tests undertaken on similar electrode materials[44,45].

Among all the battery properties, one expects on a theoretical basis that the volumetric exchange current density $ai_0^\theta$, OCP gradient $k_U$, diffusion time $t_d$, and entropy change $\Delta S$ may vary with cell SOC, while the other material properties should be nearly independent of it. In the range of SOC studied here, however, the variations of fitted $ai_0^\theta$ and $t_d$ with SOC were also minimal. The apparent constancy of $t_d$ in this range is qualitatively confirmed by the horizontal temperature-profile concavity data shown in Fig. S6, which are nearly invariant with respect to cell SOC at a given current density. For A123 20 Ah LFP pouch cells, it appears that a single, SOC-independent diffusion coefficient and exchange current density suffice to describe measured voltage and temperature behaviour at C-rates up to 4C.

For validation purposes, simulations using the fitted parameters were also compared with experimental results at conditions not used for fitting, with different C-rates and cycling periods, specifically 4C-50s and 2C-100s cycling, at each SOC. Figure S6 provides the experimental and simulated voltage and temperature responses for two validation tests at 30% SOC (2C-100s@30% and 4C-50s@30%), and two 4C-100s parameter-estimation tests at 50% and 70% SOC. Measured and predicted thermal images under various square-wave cycling profiles at $t = 2500$ s are plotted in Fig. S7. The root-mean-square errors comparing simulations with experiments are 0.2 K for temperature and 5.0 mV for voltage.

## Discussion: cross-scale effects of non-uniform temperature

The model parameterised above can, with minimal modification, be expanded to full discharge simulations that retain many parameters estimated from square-wave-excitation tests. As mentioned before, most parameters vary negligibly with SOC. Full discharges were simulated by leaving every parameter constant apart from the local OCP gradient and the entropy change, which were replaced by local functions of SOC gathered from either full-cell measurements (OCP) or manufacturer-supplied data (entropy change). A description of the model parameterisation is available in Supplemental Note 4; a complete set of parameters is given in Table S3; details of how OCP and entropy were handled are discussed in the Methods section.

The maximum C-rate explored in this work is 4C, a fairly challenging test of the accuracy and generality of our approach. The LFP pouch cell was discharged at constant current from 100% SOC to 0% SOC, after first charging the cell to 3.6 V with a 'CC-CV' protocol (in which a 4C current was applied until the voltage reached 3.6 V, at which it was held until current decayed to C/100) and then resting for an additional hour. The cell voltage, current, and surface-temperature distribution were measured at 1.0 Hz. Figure 4a, b show the battery voltage and temperature response during the 4C discharge. Initially

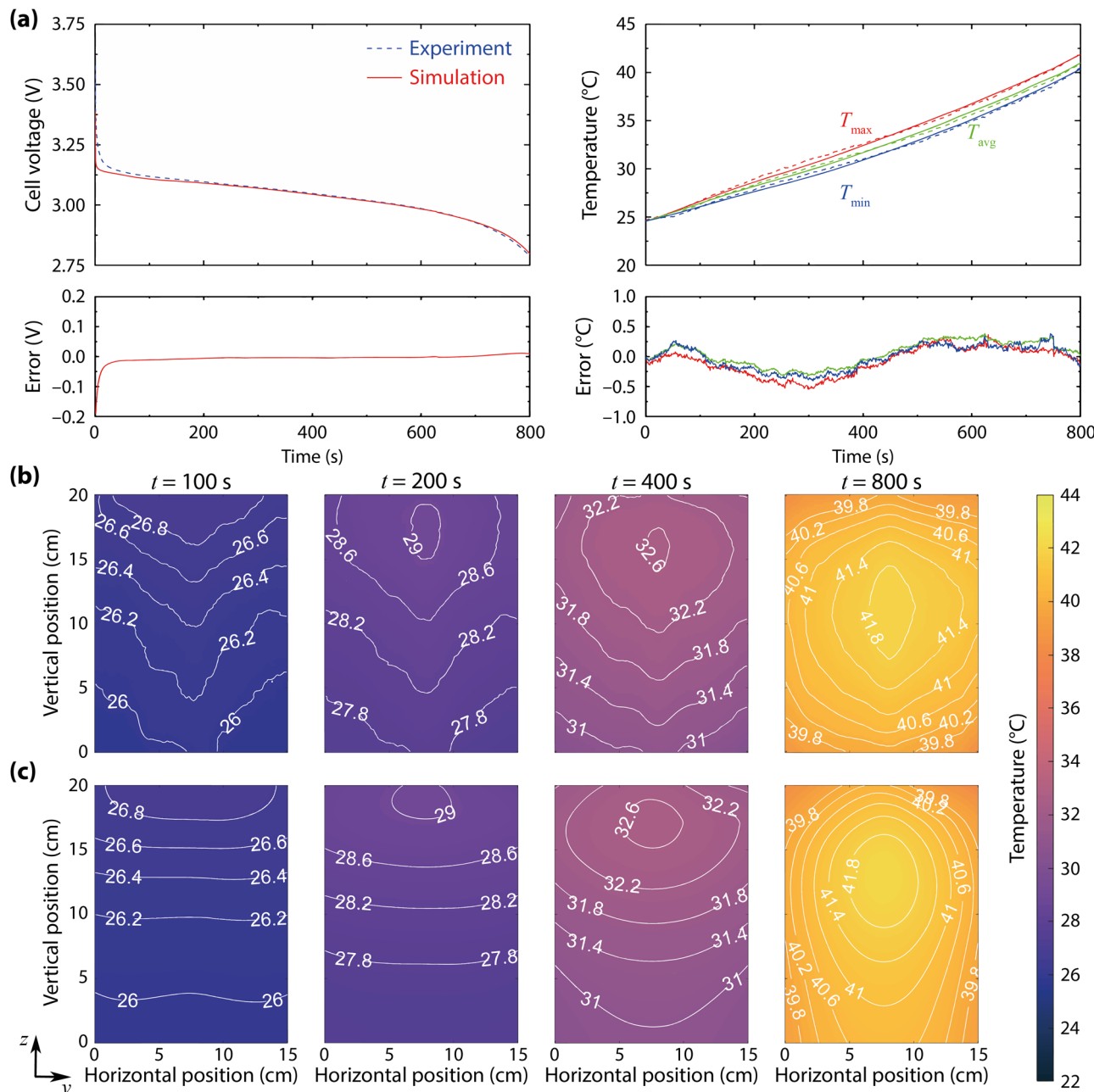

**Fig. 4 Complete constant-current discharges can be simulated by extrapolating parameters gathered from square-wave-current perturbations. a** Cell voltage and maximum ($T_{max}$), surface-average ($T_{avg}$), and minimum ($T_{min}$) temperatures during full discharge at 80 A (a 4C rate); solid lines show simulation data and dashed lines, experimental data. **b** Experimental thermograms showing the surface-temperature distribution at various times. **c** Simulation results. The battery tabs (not shown) are on the top edge of the thermal images in **b**, **c**.

($t < 100$ s), similarly to the square-wave cycling tests, the region close to the battery's tabs rapidly warms up due to the higher current density there, forming a hot spot near the cell's top edge. Between $t = 100$ s and $t = 200$ s, the size of the hottest area gradually expands vertically; the hot spot detaches from the top edge at $t = 200$ s. As the discharge continues, the hot spot moves down towards the centre of the cell and the hottest area grows in size. The hot spot shifts downward substantially, passing through *ca.* $z = 160$ mm at $t = 400$ s and $z = 110$ mm at $t = 800$ s.

The average surface temperature increases by 17 °C during the 4C discharge. At the end of discharge, the surface-temperature distribution spans 1.5 °C between the maximum temperature, at the hot spot, and the minimum temperature, at the bottom edge. Simulated transient surface-temperature fields are shown in

Fig. 4c and a real-time comparison between the thermography test and model simulation is provided in Supplementary Movie 2; the temperature rise and distribution are well matched between the model and experiment throughout the discharge.

Figure 5a shows spatial and temporal variations of solid-phase, liquid-phase, and reaction current density through-plane (i.e., normal to the $y$–$z$ plane shown in Fig. 4) through the in-plane ($y$–$z$) location of the hot spot.

Generally, the magnitude of liquid-phase current density increases toward the separator, while the solid-phase current drops. Reaction current is associated with the concavity of these profiles. Extrema of the reaction current are observed in both electrodes at all times. Electrochemical reactions in the electrodes are initially favoured at the electrode/separator interfaces, so the

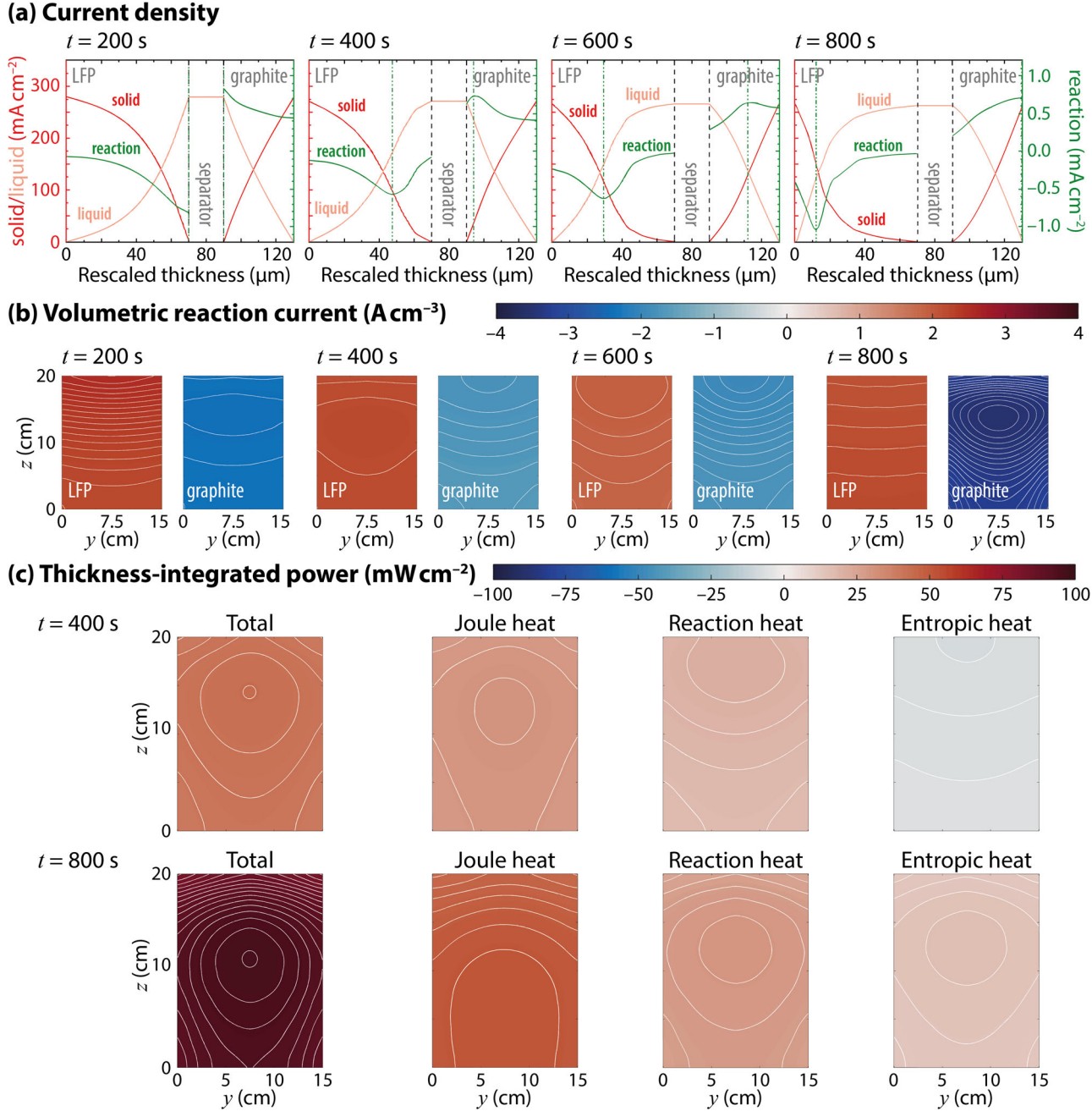

**Fig. 5 Fine-grained simulation results for a complete fast discharge. a** Spatiotemporal variations of solid-phase (red), liquid-phase (pink), and interfacial-reaction (green) current density during a 4C (80 A) discharge, along an axis through the hot spot (at horizontal position 8 cm and vertical position 14 cm). **b** Local reaction current density in central cross-sections of the LFP and graphite electrodes through the green dash-dotted lines in **a**; contour lines are spaced at intervals of 20 mA cm⁻³. **c** Total power dissipated across the entire electrode thickness, as well as contributions from Joule, reaction, and entropic heat, at $t = 400$ s and $t = 800$ s during the discharge. Contour lines are spaced at intervals of 1 mW cm⁻².

extrema originate near the separator in both electrodes. As discharge progresses, the extrema become peaks, and the reaction front moves over time toward the current collectors. The presence of peaks in reaction current owes to the solid-phase diffusion limitation, a phenomenon familiar from one-dimensional DFN models in the literature[25]. Diffusion also controls the rate at which the reaction-current peaks move towards the current collectors. Because the graphite electrode is thinner, the extremum of reaction current has reached the current collector at 800 s, whereas the LFP reaction-current distribution still exhibits a peak. But because the LFP electrode is thicker, the peak in reaction current is narrower, and has a higher magnitude; the separator

side of the LFP is fully discharged just after 400 s, and can no longer sustain reaction current.

The current densities also vary substantially in the in-plane direction. Figure 5b shows in-plane reaction-current distributions along cross-sections cut through the instantaneous extrema of reaction current (these positions are indicated with green dash-dotted lines in Fig. 5a). Peak-to-valley variation relative to the average ranges from *ca.* 2 to 20%, and differs between the two electrodes at a given instant, despite the solid-phase diffusion times being the same. When the extremum in reaction current resides at the separator or current collector, the reaction current has relatively uniform gradient top-to-bottom. Otherwise, the

*x*-location of the peak, *cf.* Fig. 5a, is also associated with a peak in the *y*–*z* plane. Again, the peak is sharper in the thicker electrode.

Figure 5c presents the in-plane distributions of instantaneous heat generation, integrated across the thickness of the whole cell after 400 and 800 s of 4C discharge. The total heat generated is broken down into contributions from Joule heat (arising from bulk resistances in Eq. (7) of the 'Methods' section), reaction heat (interfacial resistances), and entropic heat. The local maxima in total through-plane heat generation are similar to the hot-spot locations in Fig. 4. The total heat generation generally increases as the cell is discharged. Joule heating accounts for about half of the total: at 400 s it is relatively uniformly distributed in-plane; at 800 s its distribution is dominated by an increase from the top of the cell to the bottom. Reaction heat and entropic heat, on the other hand, both come to relatively sharp peaks in the *y*–*z* plane. They interfere destructively at 400 s, where entropic heat is negative, and constructively at 800 s. Nevertheless the sum of the two is always positive; reaction heat and entropy together dominate the placement of the peak in total heat generation. Although the reaction-current distributions in Fig. 5b are difficult to parse, it does appear that the average position of the extreme reaction currents correlates with the maximum in total heat generation. Thus it can be concluded that the position of the hot spot is largely controlled by non-uniform in-plane reaction current.

## Conclusions

Multiple phenomena within a battery, including electrochemical reactions, interfacial kinetics, and bulk resistance, have distinctive heat signatures. Thus, in large lithium-ion pouch cells, surface temperature can be used as an effective probe to provide microscopic understanding. Using a combination of lock-in thermography and physics-based modelling, we characterised several material properties by inverse modelling of experimental tests with square-wave applied currents. Significant in-plane temperature non-uniformity was observed, and was attributed to a balance of the heterogeneous distributions of local charge state, interfacial Joule heating, and ohmic heat generation. Importantly, solid-phase diffusion of intercalated lithium was found to have a distinctive macroscopic heat signature, leading to concavity in the temperature distribution along an axis parallel to the current collectors but perpendicular to the battery's tabs.

We showed that many inferred properties could be assumed constant over a very wide range of states of charge. A model that kept most parameters fixed but included a lookup table of pseudo-OCP data accurately simulated the cell voltage and thermal response during galvanostatic full discharges at 4C. Joule heating and reaction heat are generally comparable in magnitude; the position of the maximum temperature on the cell surface is controlled primarily by the microscopic reaction distribution and solid-phase diffusion. Since large-format lithium-ion cells are favoured for high-energy-density packs, detailed knowledge about non-uniform thermal states is critical to the understanding of battery performance and cycle life, particularly during fast charging or discharging.

## Methods

**Electrochemical testing**. All tests reported were performed using commercial 20 Ah LFP pouch cells (AMP20M1HD-A, A123 Systems). The equilibrium OCP of the cells was measured with a pseudo-OCP approach at C/25 applied current between 3.6 V (identified as 100% SOC) and 2.5 V (0% SOC). The charge capacity was determined via Coulomb counting under a CC-CV protocol at 1C until the current decayed to C/100. When a cell was set to a certain initial SOC, it was first charged to 100% SOC with a CC-CV protocol, then discharged to the required setpoint by Coulomb counting at 1C. All these tests were conducted using an automated battery test system (Series 4000, Maccor Inc.). In the lock-in thermography tests, the cells placed in the test rig were charged and discharged using a high-power bipolar power supply (BOP 10-100MG, KEPCO Inc.). Square-wave-

excitation cycling data were gathered based on initial SOCs of 30, 50, or 70%, with applied current at 2C or 4C and periods of 50 s or 100 s[16]. Four different cells were used for the 4C-100s@50% experiments. Full-discharge experiments were conducted from 100% SOC to 0% SOC with applied current at 0.2, 1, 2, and 4C.

**Pouch cell disassembly**. To determine the appropriate physical dimensions and layer structure for use in finite-element modelling of the LFP pouch cell, a cell was fully discharged to 2.5 V, then cut open using ceramic scissors and disassembled in a glove box. The thickness of each component was measured by a micrometer, and the length and width were measured with digital calipers. Photographs of single layers of the cathode and anode extracted from the pouch cell are shown in Fig. S2. The geometric parameters of the battery and its components were measured and are summarised in Table S1.

**Lock-in thermography and thermal image analysis**. Lock-in thermography was conducted following the method of Chu et al.[16] using a thermal imaging camera (A35sc, FLIR Systems). Spatiotemporal temperature data were gathered by image processing, including reference-temperature data averaged over a piece of black felt, labelled as the 'ambient reference' rectangle in the thermogram in Fig. 1. The ambient-spot temperature from thermography was calibrated against measurements from a thermocouple (±0.1 °C accuracy, Type T, Omega Engineering) placed behind a piece of felt at that location, seen to the right of the battery in the photograph in Fig. 1. The resulting baseline temperature was subtracted from each pixel of the image during data processing.

**Battery model**. Transient surface-temperature and voltage profiles were simulated using a reduced-order continuum model, which was derived from an extended Doyle–Fuller–Newman model that incorporates a local energy balance, as detailed in Supplemental Note 2. This 3D model was applied throughout the pouch interior. As described in the Results section above, a simplified model, which neglects diffusion in the electrolyte but retains solid-state diffusion in the electrodes, was deemed sufficient to fit the data.

Within the macroscopic volume spanned by a given electrode, this reduced-order model considers ohmic charge balances in the solid and liquid phases, respectively, such that

$$\nabla \cdot \vec{i}_s = -ai, \tag{2}$$

$$\nabla \cdot \vec{i}_l = ai, \tag{3}$$

$$\vec{i}_s = -\sigma \nabla \phi_s, \tag{4}$$

$$\vec{i}_l = -\kappa \nabla \phi_l, \tag{5}$$

in which $\vec{i}_k$ and $\phi_k$ are respectively the current density and electrical potential in phase $k$ (with subscript 'l' designating liquid and 's', solid), $a$ is the pore surface area per unit electrode volume, and $i$ is the current density across the pore surface, defined such that anodic currents are positive. The effective electronic conductivity of the solid is $\sigma$, and the effective ionic conductivity of the liquid, $\kappa$. Ionic conductivity was taken to vary with absolute temperature $T$ according to

$$\kappa = \kappa^\theta + \alpha(T - T^\theta), \tag{6}$$

where $\kappa^\theta$ is the conductivity at reference temperature $T^\theta$ and $\alpha$ expresses its linear variation.

Temperature is distributed across the electrode domains according to a macroscopic thermal energy balance, derived under the assumption that the interpenetrating liquid and solid phases that make up the electrode sit at equal temperatures:

$$\tilde{C}_p \frac{\partial T}{\partial t} = \nabla \cdot (k\nabla T) + \sigma \nabla \phi_s \cdot \nabla \phi_s + \kappa \nabla \phi_l \cdot \nabla \phi_l + ai\eta + aiT\Delta S. \tag{7}$$

Here $\tilde{C}_p$ is the effective local volumetric heat capacity, $k$ is the effective thermal conductivity, and $\Delta S$ is the reaction entropy of the electrode half-reaction.

The applied current ($i_{app} = \vec{n} \cdot \vec{i}_s$) and the electric ground ($\phi_s = 0$) are defined at the positive and negative terminals (copper bars, *cf.* Fig. S10), respectively. The component of $\vec{i}_l$ normal to interfaces between the current collector and the electrodes was taken to vanish; similarly, the components of $\vec{i}_s$ normal to interfaces between the anode and cathode were taken to vanish. The outer edges of the pouch were taken to be electrically insulating. To bound the thermal portion of the problem, Newton's law of cooling was adopted at the exterior surfaces of the cell:

$$-(\vec{n} \cdot k\nabla T)\big|_{t,pouch} = h\Big(T\big|_{t,pouch} - T_0\Big), \tag{8}$$

in which $h$ is the heat transfer coefficient and $T_0$ is the ambient temperature.

Active particles within each electrode are taken to exist across a microscopic radial dimension $r$ at each point within the electrode. Within these spherical particles of

radius $r_0$, the concentration $c_s$ of intercalated lithium is taken to satisfy Fick's law

$$\frac{\partial c_s}{\partial t} = \frac{D_s}{r^2} \frac{\partial}{\partial r}\left(r^2 \frac{\partial c_s}{\partial r}\right), \qquad (9)$$

in which $D_s$ is the solid-phase diffusivity. This microscopic mass balance is coupled to the macroscopic problem through a boundary condition

$$-D_s \frac{\partial c_s}{\partial r}\bigg|_{t,r_0} = \frac{i}{F}, \qquad (10)$$

where $F$ stands for Faraday's constant, and $c_s(t, 0)$ is required to be finite. Thus the local macroscopic interfacial current density $i$ completely specifies the diffusion dynamics within the solid particles.

As justified in Supplemental Note 2, the interfacial reaction currents were taken here to follow linear kinetics,

$$i = i_0 \frac{F\eta}{RT}, \qquad (11)$$

where $i_0$ is the exchange current density and $R$, the gas constant. The temperature dependence of exchange current density was modelled as[46]

$$i_0 = i_0^\theta \exp\left[-\frac{E^\theta}{R}\left(\frac{1}{T} - \frac{1}{T^\theta}\right)\right], \qquad (12)$$

in which $i_0^\theta$ is the exchange current density at $T^\theta$ and $E^\theta$ is an Arrhenius parameter. Charge transfer is driven by the surface overpotential $\eta$ between the liquid and solid phases, which breaks down as

$$\eta = \phi_s - \phi_l - U, \qquad (13)$$

in which $U$ is the electrode's equilibrium OCP relative to a reference electrode of a given kind.

During square-wave cycling tests, the perturbation in SOC is small, allowing it to be linearised following Chu et al.[16]. Whereas the equation of Chu is based on the average SOC within the particle, the diffusion limitation requires that this be replaced by the effective SOC at the particle surface, $q$, defined as

$$q = \frac{c_s|_{t,\text{surf}}}{c_{s,\text{max}}}. \qquad (14)$$

Here $c_s|_{t,\text{surf}}$ is the instantaneous surface concentration at $r = r_0$ within the solid and $c_{s,\text{max}}$ is the maximum lithium concentration the solid particles can accept. Taking account of the diffusion limitation's effect on the surface concentration of intercalated lithium, one finds that the OCP satisfies

$$U = U_0 + k_U(q - q_0) + \frac{\Delta S_0}{F}(T - T_0) + V_\text{hys} \cdot \text{sgn}(i), \qquad (15)$$

in which $q_0$ is the average fractional state of charge of the whole pouch cell. The constant parameter $V_\text{hys}$ is included to describe possible OCP hysteresis during slow charge or discharge[47]. Here, a nonzero value of $V_\text{hys}$ was included to model LFP cathodes; $V_\text{hys}$ was taken to be zero for graphite anodes. Here, $U_0$ is the OCP at $q_0$, $k_U$ represents the OCP gradient with respect to fractional SOC at $q_0$, and $\Delta S_0$ is the reaction entropy at $q_0$ and $T_0$, which appears because OCP satisfies the Maxwell relation

$$\Delta S(q) = F\left(\frac{\partial U}{\partial T}\right)_q. \qquad (16)$$

Generally we assume that the reaction entropy depends weakly on temperature, and is therefore a function of SOC only.

The same numerical model was used to simulate full discharges, except the linearised OCP curves from Eq. (15) were replaced with full-cell experimental pseudo-OCP discharge data in the nonlinear form

$$U = U_0(q, T_0) + \frac{\Delta S(q)}{F}(T - T_0), \qquad (17)$$

where the functions $U_0$ and $\Delta S$ come from experiments. (Note that no hysteresis term is present here because only discharges were modelled.) These data were used assuming the anode as a reference potential: thus the model used $U$ as the equilibrium voltage in LFP, and assumed the graphite OCP to be ground (0 V). In the model, reversible heating was computed from full-cell $\Delta S$ data by equally apportioning the reaction entropy between the two electrodes. This could be refined if reference-electrode measurements were available, but practically, the thinness of the cell normal to the electrodes means that an unequal distribution of reversible heat is difficult to discern. The full-cell OCP and $\Delta S$ data are provided in Figs. S8 and S9, respectively. A detailed description of the model parameterisation is available in Supplemental Note 4, and a complete set of model parameters is given in Table S3.

When solving the model, computational speed can be improved by a scaling analysis of the governing equations. By applying the scaling argument put forward by Chu et al.[16], the multi-layer internal geometry was homogenised, allowing the electrochemical model to be solved across a single representative layer. The scaling procedure and details of the simulated geometry are described in Supplemental Note 3.

The battery model was simulated with COMSOL Multiphysics software on a typical Windows PC (CPU: Intel Core i7, 3.0 GHz, 4 cores; 32 GB RAM). Each simulation took approximately 3–5 min, depending on the input conditions. Further speed-up could presumably be achieved by adjusting the spatial mesh and time stepping.

**Parameter estimation**. Inverse modelling was based on iterative solution of the transient battery model. The parameter vector

$$X = \left[ai_0^\theta, E_{i0}, \kappa, \alpha, k_U, \frac{r_0^2}{D_s}, \Delta S, \frac{h}{\bar{C}_p}, k\right] \qquad (18)$$

was identified using a nonlinear least-squares fitting algorithm that minimised the error between measurements and simulations (respectively denoted with superscripts 'exp' and 'sim') at each time-step $i$ of voltage, $V_i$; maximum, minimum, and surface-averaged temperatures $T_{\text{max},i}$, $T_{\text{min},i}$, and $T_{\text{avg},i}$, respectively; concavity of the temperature distribution $k_{c,i}$; and the position of the hot spot in the $yz$ plane, $(y_\text{hot}, z_\text{hot})$. The objective function $f$ for the minimisation was expressed as a sum over all $N$ entries in the time series, as

$$f = \sum_{i=1}^{N}\left[\left(\frac{V_i^\text{sim} - V_i^\text{exp}}{\Delta V^\text{exp}}\right)^2 + \left(\frac{T_{\text{max},i}^\text{sim} - T_{\text{max},i}^\text{exp}}{\Delta T_\text{max}^\text{exp}}\right)^2 + \left(\frac{T_{\text{min},i}^\text{sim} - T_{\text{min},i}^\text{exp}}{\Delta T_\text{min}^\text{exp}}\right)^2\right.$$
$$\left. + \left(\frac{T_{\text{avg},i}^\text{sim} - T_{\text{avg},i}^\text{exp}}{\Delta T_\text{avg}^\text{exp}}\right)^2 + \left(\frac{k_{c,i}^\text{sim} - k_{c,i}^\text{exp}}{\Delta k_c^\text{exp}}\right)^2 + \frac{(y_{\text{hot},i}^\text{sim} - y_{\text{hot},i}^\text{exp})^2 + (z_{\text{hot},i}^\text{sim} - z_{\text{hot},i}^\text{exp})^2}{L_y L_z}\right], \qquad (19)$$

where $\Delta V^\text{exp}$, $\Delta T_\text{max}^\text{exp}$, $\Delta T_\text{min}^\text{exp}$, $\Delta T_\text{avg}^\text{exp}$, and $\Delta k_c^\text{exp}$ denote the ranges of $V$, $T_\text{max}$, $T_\text{min}$, $T_\text{avg}$, and $k_c$ in the experiment. The width and length of the pouch cell are $L_y$ and $L_z$. The temperature concavity was calculated at the hot spot location by fitting the horizontal temperature profile through the hot spot (see Fig. S5) with a quadratic polynomial, as described in Supplemental Note 1.

## Data availability

Datasets from all lock-in thermography experiments conducted in this study are available at Oxford University Research Archive (ORA), https://doi.org/10.5287/bodleian:NGQkq49wG[48].

## Code availability

Battery models and parameter estimation codes used in this study are available at GitHub, https://github.com/Battery-Intelligence-Lab/multiscale-coupling.

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

## Acknowledgements
The authors gratefully acknowledge funding from the EPSRC Translational Energy Storage Diagnostics (TRENDs) project (EP/R020973/1), the Faraday Institution Multi-scale Modelling 2 project (subaward FIRG025 under grant EP/P003532/1) and the STFC Futures Early Career Award. The authors also thank Andrew A. Wang for his help with pouch-cell disassembly.

## Author contributions
J.L., H.N.C., D.A.H., and C.W.M. conceived the study. J.L. conducted data analysis, battery modelling, parameter estimation and drafted the initial manuscript, H.N.C. performed battery testing and lock-in thermography experiments. J.L., D.A.H., and C.W.M. revised and edited the manuscript. D.A.H. and C.W.M. supervised the work.

## Competing interests
D.A.H. is co-founder of Brill Power and is a technical adviser at Habitat Energy. All other authors declare no competing interests.
