## [Peer Review File · Communications Engineering]

Reviewers' comments:

Reviewer #1 (Remarks to the Author):

The manuscript "Multiscale coupling of surface temperature with solid diffusion in large lithium-ion pouch cells" carried out thermal modeling for the LFP battery cells, considering the in-plane current, voltage, and temperature distribution of large pouch cells. It is interesting to investigate this research, especially with the trend of higher energy density and large size. And the simulation result is improved by the model validation. However, there are still some issues that need to be addressed before any further publication:

- (1) It seems that this study is a further study based on the work by Chu et al. [16] and on the COMSOL platform, the new findings and contributions should be enhanced more in the introduction.
- (2) The writing part should be improved. Some sentences are not formal, some expressions are even too emotional. For example, why the "in-plane distribution of current is more important" as the authors mentioned in page 2, line 20? There are some supports missing? and what is the research state of the battery thermal modeling? A benchmark work is suggested.
- (3) You introduce the parameters of the battery model are introduced, but I do not find the boundary conditions, e.g., heat transfer boundary condition, and model assumptions but highly affect the simulation accuracy. More specific descriptions about the modeling process should be explained.
- (4) The reasons behinds the temperature distribution can be given, rather than simply presenting the result.
- (5) The model validation is important, the authors should conduct error analysis for the temperature and current difference between the simulation and experiments, which can be affected the laboratory environment or the precision of the sensors. And how is the model performance at extreme SOC? If possible, I would like to see more validation on a wider SOC range, which will prove the reliability of the simulation.
- (6) Since different types of Lithium-ion battery cells demonstrate different thermal or electric performance, could the conclusions be drawn for some other Lithium-ion battery?

Reviewer #2 (Remarks to the Author):

This paper presents a physics-based pouch-cell model to simulate lock-in thermography experiments

1- The paper presents a model tested and validated for one battery technology, LFP. There are previous publications already since 2015 [1] in which different battery chemistries were tested for surface temperature behavior understanding. The presented work develops a well described model and provides as well further research on microscopic intercalation processes and its consequences. However, an extension on the work for other chemistries is missing. This should be referred and described at least in the introductory part and conclusions.

2- A better comparison with the SoA should be introduced to better understand the benefits of this model.

3- It is unclear the effect on fast charging on the tested technology and model development.

4- Further explanation on the pouch cell disassembly and studies of the opening of the cell and its visual inspection referring to the developed model are missing.

5- The presented model is valid at cell level, however, would be interesting for the reader if the model can be transferred to a module level and how can this be done.

- 6- Please elaborate further on the computational effort this method requires. Can this model be embedded in a conventional micro controller?
- 7- Temperature sensor integration on battery cells is under research however costly. How can this model help the understanding on the surface temperature in relation to the number of sensors to be integrated?
- 8- It is unclear the number of cells that have been used for the model development and validation purposes. This is important for validation purposes of the model.
- 9- A link to the possible degradation mechanisms or State-of-Health estimation would be suitable to be integrated.
- 10- Section 2 should provide a clear flow chart additionally to the written explanation of the testing procedure.

[1] S. Goutam, J.-M. Timmermans, N. Omar, P. Bossche, J. Van Mierlo, Comparative Study of Surface Temperature Behavior of Commercial Li-Ion Pouch Cells of Different Chemistries and Capacities by Infrared Thermography, *Energies*. 8 (2015) 8175–8192.

Reviewer #3 (Remarks to the Author):

This work tried to model the surface temperature of Li-ion pouch cell by adding a solid diffusion term in the physics based model, an extension of streamline model developed before. The model results seem to be consistent with the experimental testing results. However, there are a couple of concerns regarding the methodology and conclusion of this paper:

- 1) The effect of solid diffusion on thermal performance of batteries depends on the setting of other properties of batteries. The effective conductivity was decreased from 50-60 (W/mK) (streamline model) to 1 (W/mK). This decreases the thermal transport. It is easier to create thermal concavity. The question is that why adding the solid diffusion term decreases the effective thermal conductivity so much? The local reaction current decreases from 143 (streamline model) to 1.86, how can adding the solid diffusion term affect the local current density?
- 2) Is it possible that the temperature concavity are affected by the non-isotropic thermal properties?
- 3) How's the local current density distribution? The temperature distribution should be consistent with the local current density distribution.

Title: Multiscale coupling of temperature non-uniformity and solid-state diffusion in lithium-ion pouch cells

Ref: Communications Engineering COMMSENG-21-0031-T

Thank you for collecting the reviews of our paper. We appreciate the helpful questions raised by the reviewers. Below are our responses, with reviewer comments indicated in blue italics. Text that has been modified or added to the manuscript is highlighted in red.

Reviewer 1

The manuscript “Multiscale coupling of surface temperature with solid diffusion in large lithium-ion pouch cells” carried out thermal modeling for the LFP battery cells, considering the in-plane current, voltage, and temperature distribution of large pouch cells. It is interesting to investigate this research, especially with the trend of higher energy density and large size. And the simulation result is improved by the model validation.

1. It seems that this study is a further study based on the work by Chu et al. [16] and on the COMSOL platform, the new findings and contributions should be enhanced more in the introduction.

It is true that the present study builds upon earlier work – namely the “streamlined model” by Chu et al. [16], but the earlier work did not include solid diffusion; this pseudo-fourth spatial dimension of the model is what leads to most of the key results we present here. In particular, solid diffusion is the underlying mechanism that we ultimately find to determine the lateral surface-temperature distribution. Also, the prior work did not put the reduced-order model in appropriate context with existing physical models; the formal derivation from Doyle–Fuller–Newman theory provided here encompasses both the present model (with solid diffusion) and the prior streamlined model.

To highlight the novelty more clearly, we have added new text to the Introduction on lines 18-29 (see response to item 2 below).

2. The writing part should be improved. Some sentences are not formal, some expressions are even too emotional. For example, why the “in-plane distribution of current is more important” as the authors mentioned in page 2, line 20? There are some supports missing? and what is the research state of the battery thermal modeling? A benchmark work is suggested.

Regarding the first point about quality of the writing: We are confident that our paper is written clearly, concisely, and dispassionately in line with the Editorial Guidelines of this journal. The text quoted by the referee appears in the Introduction section; it summarizes a point made in the literature. We cite three references, all of which assert that the ‘in-plane’ current distribution can be as or more important – in the sense that it has a dominating effect on experimental output – than the ‘through-plane’ current distribution. We do not think that when used in this context, the words “more important” are informal or amount to an unsubstantiated value judgement. We have left the tone of the text elsewhere largely unchanged. Should the copy editors at *Communications Engineering* recommend stylistic or grammatical changes in the manuscript, however, we shall be happy to incorporate them.

Regarding the comment about the state of research: We think that the reviewer makes an excellent point, and have consequently added some additional literature review (with citations) to the Introduction, clarifying the text and adding some references to address the current state of coupled thermal modelling on lines 18-29:

Thermal models are typically coupled to electrochemical models through an energy balance equation that accounts for joule (ohmic) heating, reaction heat, and entropic heat. Thermal effects were first considered in a zero-dimensional global form by Bernardi et al. [5] and later treated in a local but spatially one-dimensional form by Srinivasan and Wang [26]. Electrochemical–thermal battery

models have been modified for various cell geometries and operating conditions [13,27–29]. Most simulations deal with three-dimensional geometries by decoupling the thermal problem from the charge-transport problem, solving a one-dimensional Doyle–Fuller–Newman model normal to the electrode sandwich at a given location, then using the results of that computation to produce a generation term in a homogenized, three-dimensional heat equation [30]. Despite the fact that almost all of the porous-electrode-theory investigations in the literature focus only on the electrochemical response of a single layer in the ‘through-plane’ direction perpendicular to the current collectors [31–36], the ‘in-plane’ distribution of current can be equally or perhaps even more important, especially in large-format cells [8,18,37,38].

3. You introduce the parameters of the battery model are introduced, but I do not find the boundary conditions, e.g., heat transfer boundary condition, and model assumptions but highly affect the simulation accuracy. More specific descriptions about the modeling process should be explained.

A very detailed description of the battery model has been provided in the “Battery model” subsection of the Methods section, which has been put after the Conclusions in accordance with this journal’s format. The boundary conditions on all relevant interfaces are described in the text surrounding equation 8 (which is itself the exterior thermal boundary condition).

We nevertheless appreciate the reviewer’s point that more detail about model implementation would be useful to readers. Therefore, we have posted the COMSOL and Matlab scripts in an online database. This is now referenced on lines 368-369:

The files used to implement the model and parameter estimation are on GitHub at <https://github.com/Battery-Intelligence-Lab/multiscale-coupling>.

4. The reasons behinds the temperature distribution can be given, rather than simply presenting the result.

The discussions around Figures 2 and 5 explore and clarify the mechanisms that contribute to the observed temperature distribution. We agree that our mechanistic reasoning could be made clearer, however. We have amended the text on lines 99-102 to read:

At diffusion time constants above 100 s — values more representative of real electrode materials [40, 41] — significant extra heat generation occurs, causing an even larger temperature difference between the central vertical axis of the cell surface and its left or right edges, with generally higher absolute temperatures everywhere.

5. The model validation is important, the authors should conduct error analysis for the temperature and current difference between the simulation and experiments, which can be affected the laboratory environment or the precision of the sensors. And how is the model performance at extreme SOC? If possible, I would like to see more validation on a wider SOC range, which will prove the reliability of the simulation.

On the point about simulation error: The simulation error is addressed at the end of section 2, lines 174-176: “The root-mean-square errors comparing simulations with experiments are 0.2 K for temperature and 5.0 mV for voltage.” Error is also quantified in two of the graphs on Fig. 4a and the absolute accuracies of the sensors are discussed in the Methods section on line 287.

Note that we have added new material to the revised manuscript about cell-to-cell variability, as well. This is discussed below in the response to referee 2 (item 8).

On the point about extreme SOC: Figure 4 summarizes a full-discharge experiment and a corresponding simulation based on parameters estimated using square-wave pulse data. We believe that this provides

the validation test suggested by the referee, since it extrapolates parameters measured at fixed, moderate SOC to an experiment that runs across the entire available SOC range, including the extremes near 0% and 100% SOC. This experiment is also at 80 A (a 4C discharge) – a relatively demanding condition for the model.

6. Since different types of Lithium-ion battery cells demonstrate different thermal or electric performance, could the conclusions be drawn for some other Lithium-ion battery?

We appreciate this question, which could pertain either to cell geometry or cell chemistry.

Although the infrared imaging technique performed here is not directly applicable to cylindrical cells, the coupling between solid diffusion and heat flow likely still affects device response. Notably, temperature variation in both axial and radial directions has been observed in 18650 cells operated at high current [for example, see R. Richardson et al., *J. Power Sources* **327** (2016) 726-735].

For cells with different chemistry, we believe the conclusions will also hold. At the 17th MODVAL symposium in 2021, we presented experimental results for 16 Ah nickel manganese cobalt (NMC) pouch cells. Similar thermal behaviour was observed experimentally, and more importantly, the model predictions were similarly accurate, and the parameter estimates were similarly reproducible across cells. These data comprise part of a longitudinal study, which we have yet to report because it relies on the parametrization techniques developed in this paper.

Reviewer 2

1. The paper presents a model tested and validated for one battery technology, LFP. There are previous publications already since 2015 [1] in which different battery chemistries were tested for surface temperature behavior understanding. The presented work develops a well described model and provides as well further research on microscopic intercalation processes and its consequences. However, an extension on the work for other chemistries is missing. This should be referred and described at least in the introductory part and conclusions.

*[1] S. Goutam, J.-M. Timmermans, N. Omar, P. Bossche, J. Van Mierlo, Comparative Study of Surface Temperature Behavior of Commercial Li-Ion Pouch Cells of Different Chemistries and Capacities by Infrared Thermography, *Energies*. 8 (2015) 8175–8192.*

We thank the reviewer for this insightful comment. We have reviewed the suggested publication and are interested in the thermal model that appears there, although in the electrochemical part it differs from the well-accepted Doyle–Fuller–Newman framework (the governing equations appear to include an anode and a cathode charge balance at every interior spatial grid point, with no explicit electrolyte phase included). This difference from standard approaches makes it difficult to connect those results to the existing literature on physics-based electrochemical battery modelling. Nevertheless, this is a significant experimental reference and we have included a citation to it (number [18]) in the Introduction, after the last sentence of new text included in the response to Reviewer 1, item 2.

Regarding applicability to different chemistries, as mentioned in the response to Reviewer 1, item 6, we have observed the effects reported here in pouch cells with NMC cathodes as well as LFP. These experiments are still underway and will be reported in a later publication.

2. A better comparison with the SoA should be introduced to better understand the benefits of this model.

A literature review of battery electrochemical-thermal model has been provided in the Introduction to clarify the current state of battery thermal modelling and the benefits of the present model. Hopefully the changes given in response to the previous point and to Reviewer 1, item 2 suffice to address this concern.

3. It is unclear the effect on fast charging on the tested technology and model development.

Although we do not particularly focus on fast charging in this study, we have tested the cells up to 4C under square-wave-excitation cycling (charging and discharging) for model parameterisation, and at 4C under full discharge for model validation. Given that these cells have 20 Ah capacity, the C-rate tested corresponds to 80 A, falling within the fast-charging range.

At C-rates higher than 4C, temperature non-uniformity within the cell will be greater, making it worth the effort to develop thermal diagnostics further. It is a good question what the rate limitation of the proposed reduced-order model would be, owing to several assumptions we have made, such as infinitely fast diffusion in the electrolyte phase. This question is currently under investigation in our laboratory, where we are pushing both experiments and modelling up to 10C to understand high-power cells.

It is notable that the model we present derives formally from the Doyle–Fuller–Newman model, so it is always possible to relax some assumptions to account for new effects expected at very high C-rates.

4. Further explanation on the pouch cell disassembly and studies of the opening of the cell and its visual inspection referring to the developed model are missing.

The purpose of pouch-cell disassembly was to gather correct information about the battery geometry, including the layer structure and physical dimensions of each cell component (cathode, anode, separator, and current collector). These geometric parameters are needed within the model. The subsection on pouch-cell disassembly in the Methods section (lines 276-281) has been expanded and edited for clarity:

To determine the appropriate physical dimensions and layer structure for use in finite-element modelling of the LFP pouch cell, a cell was fully discharged to 2.5 V, then cut open using ceramic scissors and disassembled in a glove box. The thickness of each component was measured by a micrometer, and the length and width were measured with digital calipers. Photographs of single layers of the cathode and anode extracted from the pouch cell are shown in Figure S1. The geometric parameters of the battery and its components were measured and are summarized in Table S1.

Note that Figure S2 of the supplementary information supplies a photograph of single cathode and anode layers extracted from the pouch cell.

5. The presented model is valid at cell level, however, would be interesting for the reader if the model can be transferred to a module level and how can this be done.

The cell-level model can be easily exported as a submodel, which can be called via MATLAB. Hence, multiple cells can be used to simulate a module with appropriate adjustments of the thermal boundary conditions and electrical configuration. This is an important idea but is clearly beyond the scope of the present study.

6. Please elaborate further on the computational effort this method requires. Can this model be embedded in a conventional micro controller?

Thank you for this query about computational expense. The following text has been added at the end of the Battery Model subsection of the Methods section (lines 365-368):

The battery model was simulated with COMSOL Multiphysics software on a typical Windows PC (CPU: Intel Core i7, 3.0 GHz, 4 cores; 32 GB RAM). Each simulation took approximately 3-5 minutes, depending on the input conditions. Further speed-up could presumably be achieved by adjusting the spatial mesh and time stepping.

Embedding of the model in a microcontroller is beyond the scope of the present research but could be

an interesting control engineering exercise.

7. Temperature sensor integration on battery cells is under research however costly. How can this model help the understanding on the surface temperature in relation to the number of sensors to be integrated?

The answer to this probative question is not immediately obvious. Our key objective was to parametrize and a parsimonious physics-based model to predict the surface-temperature and charge distribution in a pouch cell. Thus, our primary focus was to extract values of the model parameters directly from experimental data, as summarized in Table 1.

We do also show, however, that for a given battery cell the parameters gathered under one operational scheme (relatively short-period square-wave cycling) can be used to predict the electro-thermal response in another scheme (constant-current full discharge) with good accuracy. Thus, once suitably parametrized, the model is predictive. In principle, there should be no problem with using the parametrized cell-level model to perform experiments *in silico* for optimization of sensor configurations. Of course, the thermal conditions would be substantially different within a module or pack, so eq. 8 of the manuscript would need reframing.

8. It is unclear the number of cells that have been used for the model development and validation purposes. This is important for validation purposes of the model.

This is a very important point, and we thank the referee for noticing our significant omission. In fact we processed lock-in thermography data from four different cells to probe cell-to-cell variation in both the observed voltage/surface-temperature responses and the extracted parameters. Variation was generally minimal.

Since these experiments are labour-intensive to set up, we believe that the raw data could be useful to the broader research community. Therefore, we have created an online data archive (doi: 10.5287/bodleian:NGQkq49wG) with all the data sets we gathered from all four cells.

This is a huge amount of data to parse, however. We therefore performed some additional data processing to quantify the extent of cell-to-cell variability more clearly. A new table (Table S2 of the supplementary information) has been added to summarize the parameters gathered from 4C-100s@50% square-wave tests on four different cells. These show very similar values for all properties between cells – most agree with standard error of 1% from cell to cell; the most variable properties are OCP gradient, which varies by 8%, ionic conductivity (7%), and diffusion time (4%). This is an excellent level of agreement.

We have added the following text to the manuscript on lines 126-132:

Replicate square-wave excitation tests were also performed for three other cells from the same manufacturing lot. Raw data for all the lock-in thermography experiments are available in a repository at ORA (doi:10.5287/bodleian:NGQkq49wG). Estimated property values from all four cells tested are reported in Table S2. Cell-to-cell variation in the parameters extracted from model fitting was explored using the 4C-100s excitation tests around 50% SOC. Parameter estimates agreed well across cells. The most variable property estimates were OCP gradient, with a standard error of 8%, ionic conductivity (7%), and diffusion time (4%); all other parameters agreed within 1% from cell to cell.

9. A link to the possible degradation mechanisms or State-of-Health estimation would be suitable to be integrated.

Indeed. This will be the basis for our next publication.

10. Section 2 should provide a clear flow chart additionally to the written explanation of the testing procedure.

We have added the requested flow chart to the supplementary information as **Figure S1**.

Reviewer 3

This work tried to model the surface temperature of Li-ion pouch cell by adding a solid diffusion term in the physics based model, an extension of streamline model developed before. The model results seem to be consistent with the experimental testing results. However, there are a couple of concerns regarding the methodology and conclusion of this paper:

1. The effect of solid diffusion on thermal performance of batteries depends on the setting of other properties of batteries. The effective conductivity was decreased from 50-60 (W/mK) (streamline model) to 1 (W/mK). This decreases the thermal transport. It is easier to create thermal concavity. The question is that why adding the solid diffusion term decreases the effective thermal conductivity so much? The local reaction current decreases from 143 (streamline model) to 1.86, how can adding the solid diffusion term affect the local current density?

The questions raised here highlight some of the key problems we aimed to address in this paper.

Regarding the conductivity: As discussed on p. 9 lines 151 ff, in the previous streamlined model, the solid-phase diffusion was assumed to be fast (this limit of the new model is shown at the left of the plot on Fig. 2), and the in-plane distribution of reaction current induced by diffusional resistance – which in turn impacts the temperature distribution – was thereby omitted. As discussed in the same place, thermal conductivity also impacts the hot-spot location. These two factors cause conductivities in the streamlined model to be dramatically overestimated. Notably the new 1 W/mK value for thermal conductivity obtained in this work is in line with literature.

Regarding the reaction current: We apologize that our nomenclature was somewhat garbled. Properly the variable i_0 is an ‘exchange current density’. We have corrected references that erroneously called it “local reaction current” in Table 1 and in the main text. Exchange current density is a constant that parametrizes interfacial reaction kinetics, and inverse exchange current corresponds to interfacial resistance. Once our error in nomenclature has been resolved, it should be clear that the text already addresses the referee’s question on p. 9 lines 153-155: “Reduced ionic conductivity [in the streamlined model] increases the amount of Joule heating, causing an overestimation of exchange current density in order to lower interfacial resistance and match the temperature.”

2. Is it possible that the temperature concavity are affected by the non-isotropic thermal properties?

Great question. This is indeed the first step we took when trying to address the inconsistency between the experiments and the streamlined model of Chu et al. Numerical experiments showed that manipulating the thermal properties, especially introducing anisotropic thermal conductivity, cannot produce the temperature concavity seen in experiments. Note also that we have already conducted another study (available as a preprint at <https://arxiv.org/abs/2112.09768>) to measure the three-dimensional anisotropic thermal conductivity of the cells studied in this paper. This study shows, as expected, that the in-plane conductivity is much higher than the through-plane conductivity because the electrode sandwich exhibits two different principal axes of a laminar composite (through-plane the high-conductivity current collectors and low-conductivity active materials are thermal resistances in series; in-plane they are in parallel).

3. How's the local current density distribution? The temperature distribution should be consistent with the local current density distribution.

All the components of current density along an axis through the hot spot are shown directly in Figure 5a. The in-plane distribution of reaction current density is shown in Figure 5b. The in-plane variation of liquid-phase and solid-phase current density can be inferred from the Joule heat panel of Figure 5c. The resulting transient temperature distribution is, by design, consistent with the heat generation – although there is a transient aspect since the thermal and electrical timescales differ greatly (*cf.* eq. 1).

REVIEWERS' COMMENTS:

Reviewer #1 (Remarks to the Author):

Thanks for the author's revision letter and most questions are well answered in the revised version. I still have the questions as follows:

Because it is the highlighted work to study the "in-plane" temperature distribution in the large-format cells, my colleague still thinks that the importance (or more importance) of the "in-plane" temperature should be explained more, not just given the literature.

We have a quick look at the refs 8,18,37,38, they are almost on the experiments concerning lithium plating, fast charging and etc. We still think that a clear statement for the advances of this study should be given. Is this study the first study to model the "in-plane" temperature? Or is it the first study of the coupling of surface temperature with the solid diffusion?

Reviewer #2 (Remarks to the Author):

The comments of the reviewers have been considered and the manuscript has been accordingly modified. The responses to the reviewers are well justified.

From the response 8 of reviewer 2, it is understood that 4 (+3) cells were used for this study. It is important to highlight the number of tested cells along the manuscript, like in the abstract and conclusions. Additionally, the fact of using 4 (+3) cells brings a question related to the robustness, reproducibility, and validation of the model. It would be suitable to integrate some comments on this in the manuscript.

Reviewer #3 (Remarks to the Author):

The authors have addressed my concerns in the revised manuscript.

Title: Multiscale coupling of temperature non-uniformity and solid-state diffusion in lithium-ion pouch cells

Ref: Communications Engineering COMMS-21-0031-T

Dear Editor,

We appreciate the further comments brought by the reviewers. Below are our responses, with reviewer comments indicated in blue italics. Text that has been modified or added to the manuscript is highlighted in red.

Reviewer 1

Thanks for the author's revision letter and most questions are well answered in the revised version. I still have the questions as follows:

1. Because it is the highlighted work to study the "in-plane" temperature distribution in the large-format cells, my colleague still thinks that the importance (or more importance) of the "in-plane" temperature should be explained more, not just given the literature.

To explain this further, on lines 31-35 we have added the following sentences:

Practically, both safety and degradation are impacted by temperature nonuniformity. Local hot spots typically have lower resistance than their surroundings, causing the active material there to be stressed more intensively by cycling. Moreover, although the global temperature of a cell may be within safe operating limits, catastrophic failure due to thermal runaway can be induced if this limit is exceeded locally.

2. We have a quick look at the refs 8,18,37,38, they are almost on the experiments concerning lithium plating, fast charging and etc. We still think that a clear statement for the advances of this study should be given. Is this study the first study to model the "in-plane" temperature? Or is it the first study of the coupling of surface temperature with the solid diffusion?

The advance in this work is indeed to explore the coupling of surface temperature with solid diffusion. To our knowledge this not been explored in the past, either in the prior streamlined model of Chu et al. or the other references listed in the referee's comment.

With help from the journal editor, we have rewritten the abstract to make our key contribution clearer:

Untangling the relationship between reactions, mass transfer, and temperature within lithium-ion batteries enables approaches to mitigate thermal hot spots and slow degradation. Here, we develop an efficient physics-based three-dimensional model to simulate lock-in thermography experiments, which synchronously record the applied current, cell voltage, and surface-temperature distribution from commercial lithium iron phosphate pouch cells. We extend an earlier streamlined model based on the popular Doyle–Fuller–Newman theory, augmented by a local heat balance. The experimental data reveal significant in-plane temperature non-uniformity during battery charging and discharging, which we rationalize with a multiscale coupling between heat flow and solid-state diffusion, in particular microscopic lithium intercalation within the electrodes. Simulations are exploited to quantify properties, which we validate against a fast full-discharge experiment. Our work suggests the possibility that non-uniform thermal states could offer a window into – and a diagnostic tool for – the microscopic processes underlying battery performance and cycle life.

We have also added a title to the caption of Figure 2, emphasizing this message:

Horizontal temperature concavity through the hot spot is controlled by solid-state diffusion time.

Reviewer 2

The comments of the reviewers have been considered and the manuscript has been accordingly modified. The responses to the reviewers are well justified.

We appreciate this favourable assessment.

1. From the response 8 of reviewer 2, it is understood that 4 (+3) cells were used for this study. It is important to highlight the number of tested cells along the manuscript, like in the abstract and conclusions. Additionally, the fact of using 4 (+3) cells brings a question related to the robustness, reproducibility, and validation of the model. It would be suitable to integrate some comments on this in the manuscript.

First, a point of clarification – we studied 4 cells, not 4 (+3).

We have already highlighted the number of cells tested in lines 132-137 and line 278, where the reproducibility and uncertainty of the cell properties parameterized using the same method are discussed.

On lines 40 and 278 we modified the text to emphasize the cases where **four** cells were studied using lock-in thermography.

Last, we should like to draw the reviewer's attention to the excellent degree of reproducibility of parameter estimates across the four cells, as highlighted on lines 135-138 already:

“Cell-to-cell variation in the parameters extracted from model fitting was explored using the 4C-100s excitation tests around 50% SOC. Parameter estimates agreed well across cells. The most variable property estimates were OCP gradient, with a standard error of 8%, ionic conductivity (7%), and diffusion time (4%); all other parameters agreed within 1% from cell to cell.”

Editorial comments

1. Would it make more sense to move the temp scale in Figure 1 to the left hand side?

Note that we have substantially changed the layouts of all five Figures to bring them in line with journal requirements regarding size, resolution, etc. When doing this we completely re-designed Figure 1 to make it a single-column figure without an inset. Now the temperature scale appears as a legend, under the thermogram on the right.

2. Figure S2 needs a scale bar. And note all supplementary Figures, tables and notes need to be referred to in the main text.

We added a scale bar to Figure S2 and made sure all supplementary figures, tables and notes have been referenced in the main text.

3. We see that the videos did not transfer to the A-version of this paper. Please ensure they are available in the next version.

The file sizes of the prior videos were too large. We have made smaller files, which hopefully will upload successfully with this revision.